# Association between childhood maltreatment, psychopathology and DNA methylation of genes involved in stress regulation: Evidence from a study in Borderline Personality Disorder

**Vera Flasbeck, Martin Brüne** *

Division of Social Neuropsychiatry and Evolutionary Medicine, LWL University Hospital Department of Psychiatry, Psychotherapy and Preventive Medicine, Ruhr-University Bochum, Bochum, Germany

* martin.bruene@rub.de

**Data Availability Statement:** All relevant data are within the manuscript and its Supporting information files.

## Abstract

Previous research suggests that childhood maltreatment is associated with epigenetic modification of genes involved in hypothalamic-pituitary-adrenal (HPA) functioning, which could cause dysregulation of the stress response system. If pervasive, this may be associated with the development of stress-related disorder in adults, including affective disorders, anxiety disorders, post-traumatic stress disorder (PTSD) or borderline-personality disorder (BPD). The majority of studies have focused on DNA methylation of the glucocorticoid receptor gene (NR3C1) and the FKBP5 encoding gene, which regulates the sensitivity of the glucocorticoid receptor (GR). How methylation of NR3C1 and FKBP5 interferes with childhood adversity and psychopathology as well as empathy is an under-researched issue. Here, we sought to investigate the association of childhood maltreatment in a sample of 89 individuals (44 healthy participants and 45 patients diagnosed with BPD) with the methylation of the $1_F$ promoter region of NR3C1 and the intron 7 of FKBP5 as well as with different measures of psychopathology and empathy. Methylation of FKBP5 (bin 2) correlated with anxiety (SCL-90-R) and the global psychopathological symptom load index (GSI), as well as with lower empathic perspective-taking abilities. Psychopathology and empathy impairments correlated with the level of childhood maltreatment. No difference in FKBP5 methylation was observed between the clinical and the non-clinical group. Methylation of NR3C1 was lower in BPD patients compared to controls, yet with small differences. The results are discussed regarding their biological relevance, including possible evolutionary explanations. In short, the regulation of the GR sensitivity by methylation of FKBP5 correlated with psychopathology and empathy scores, while no correlation emerged with the severity of childhood adversity.

**Funding:** The author(s) received no specific funding for this work.

**Competing interests:** The authors have declared that no competing interests exist.

## Introduction

Abundant research has demonstrated the role of negative life events, including traumatic childhood experiences for the development of stress-related psychiatric disorders [1]. Neuro-physiologically, childhood maltreatment is known to affect the hypothalamic-pituitary-adrenal (HPA) stress response, with newer research suggesting that epigenetic alteration (e.g., methylation or histone acetylation) of genes involved in the regulation of steroid turnover is critically involved [2,3]. The most extensively investigated genetic loci in this regard comprise the gluco-corticoid receptor gene (NR3C1) and the FKBP5 gene. The NR3C1 gene encodes for the gluco-corticoid receptor (GR), which is the main binding site of cortisol. Cortisol is released from the adrenal glands upon adreno-corticotropic hormone (ACTH) secretion in response to stressful experiences. FKBP5 modulates the sensitivity of the GR [4], whereby the FKBP5 cod-ing gene is in turn influenced by environmental stressors [5]. The GR translocation from the cytoplasm to the nucleus is controlled by the heatshock proteins (Hsp) 70 and 90, p23 and FK506-binding proteins FKBP 51 and 52 [6,7]. FKBP 52 is a co-chaperone for Hsp 90 which reduces translocation of the GR-complex [8–10]. Physiologically, activation of the GR leads to fast FKBP5 induction and binding to the GR-complex, which results in a decrease in cortisol affinity of the GR and reduced translocation to the nucleus. Thus, FKBP5 modulates the sensi-tivity of the GR and is suggested to act as a short negative feedback loop [4,5,11,12].

A disruption of this feedback loop is suggested to be a causal mechanism of the develop-ment of stress-related disorders. For example, previous research reported partial glucocorti-coid resistance in mood disorders and post-traumatic stress disorder (PTSD) due to an impaired GR signaling and regulation loop [13,14]. In addition, since traumatic events during childhood are frequently related to psychopathology in adulthood (for reviews see [1,15]), exploring the role of epigenetic modification of stress-regulating genes may help to decipher the underlying mechanism.

In recent years, both hypermethylation and hypomethylation of the NR3C1 and FKBP5 were found to be associated with childhood maltreatment [16–20]. Altered methylation of these genes was also found in patients with stress-related psychiatric disorders including mood disorders, PTSD, and borderline personality disorder (BPD; [21–25], for reviews [26–28]). Similar inconclusive findings have been reported for generalized anxiety disorder (GAD; [29,30]), with some preliminary research suggesting that cognitive behavior therapy may affect methylation of these genes [31,32].

Research on epigenetic alterations has several methodological limitations that impede com-parability of studies. As regards NR3C1, the CpG sites investigated varied across studies, even though the exon $1_F$ promoter region is the most-studied region of this gene. However, depend-ing on the CpG site, previous studies reported hypo-or hypermethylation in association with childhood maltreatment (for a review, see [27]). A similar problem arises for the study of FKBP5. For example, Tyrka and colleagues investigated two CpGs in intron 7 and reported lower levels of methylation in maltreated children compared to non-maltreated children [33]. Importantly, polymorphic variation of the FKBP5 seems to be involved, as the association of childhood maltreatment with decreased FKBP5 (intron 7) and increased NR3C1 ($1_F$) methyla-tion was moderated by the FKBP5 SNP rs1360780 [19]. In contrast to studies linking methyla-tion patterns of genes involved in stress regulation to childhood maltreatment, there is a dearth of research exploring how this may relate to general psychopathology and empathy, especially for FKBP5. Accordingly, the present study aimed to explore the methylation patterns of NR3C1 ($1_F$ promoter) and FKBP5 (intron 7), including potential correlations with child-hood maltreatment and other behavioral measures in relation to stress in a sample of individu-als who were psychologically healthy and a sample of patients with BPD. This clinical group

was chosen, because it is well known that many have experienced childhood trauma, and have difficulties in empathizing with self and others [34,35]. We hypothesized changes in methylation levels of both genes in BPD. In addition, we expected correlations of methylation patterns with the severity of psychopathological symptoms, childhood maltreatment as well as impaired empathy.

## Materials and methods

### Participants

For the current study we recruited 45 female in-patients with Borderline Personality Disorder (BPD) from the LWL-University Hospital Bochum, diagnosed according to DSM-5 criteria and a structured interview [36]. We further recruited 44 female healthy control participants via advertisement, with no history of any psychiatric condition. The age of participants was between 18 and 50 years. The group of BPD patients was unaffected by severe somatic disorders and neurological illness. Pregnancy was also excluded. Table 1 summarizes the comorbid psychiatric disorders and medication of the BPD patients. The study was approved by the Ethics Committee of the Medical Faculty of the Ruhr-University Bochum (project number 4639–13). The authors assert that all procedures contributing to this work comply with the ethical standards of the relevant national and institutional committees on human experimentation and with the Helsinki Declaration of 1975, as revised in 2008. All participants gave their full informed consent in writing.

### Questionnaires

Subjective empathy was measures using a German version of the Interpersonal Reactivity Index [37], called "Saarbrücker Persönlichkeits-Fragebogen" [38]. This questionnaire comprises four sub-scores, namely "perspective taking" (PT), "fantasy" (FS), "'empathic concern" (EC) and "personal distress" (PD) and has a good reliability with a Cronbach's alpha of 0.78.

In addition, the short German version of the Childhood Trauma Questionnaire (CTQ) was used to assess the experience of maltreatment during childhood. The questionnaire comprises 28 questions tapping into the history of emotional abuse, physical abuse, sexual abuse, emotional neglect and physical neglect. Participants were asked to rate the occurrence of maltreatment on a 5-point Likert scale (1 = never, 5 = very often). Cronbach's alpha values for the German version were high for all subscales (0.80), except for the physical neglect scale [39].

**Table 1. Comorbid disorders and medication of patients with BPD in absolute (n) and relative (in %) amounts.**

|  | n | % |
| --- | --- | --- |
| **Comorbid disorders of patients with BPD** |  |  |
| Depressive episode | 23 | 51.1 |
| Posttraumatic stress disorder | 8 | 17.8 |
| Phobic/anxiety disorder | 2 | 4.4 |
| Substance misuse | 13 | 28.9 |
| **Medication** |  |  |
| Without regular medication | 17 | 37.8 |
| Antidepressant | 18 | 40.0 |
| Antipsychotic | 2 | 4.4 |
| Antidepressant and antipsychotic drugs | 8 | 17.8 |
| Anticonvulsiva | 2 | 4.4 |
| Other psychoactive drugs | 1 | 2.2 |

In order to examine general psychological symptoms, participants were asked to complete the German Version of the Symptom-Checklist SCL-90-R Questionnaire [40,41]. Participants were required to rate 90 items on a 5-point Likert scale (0 = Not at all to 4 = severe). Based on the 90 items, values for the following nine SCL-90-R sub-scales were calculated: Somatization, Obsessive Compulsiveness, Social insecurity, Depression, Anxiety, Aggression, Phobic Anxiety, Paranoid thinking, and Psychoticism. In addition, three global indices, the Global Severity Index (GSI), the Positive Symptom Total (PST) and the Positive Symptom Distress Index (PSDI) provided information of the level of overall psychological distress.

The raw values of all scores were transformed into T values, which take sociodemographic factors into consideration. Therefore, T-values of 60 or more are regarded as mildly increased, of 70 or more as greatly increased, and of 75 or more as very greatly increased. Internal consistency (Cronbach's alpha) ranges from $r = 0.75$ to $r = 0.87$ [41].

## Epigenetic analyses

The DNA samples were collected using mouthwash (Listerine) and Oragene OG-500 collection kits (DNA Genotek, Inc., Ottawa, ON, Canada). For mouthwash samples, participants were asked to rinse the mouth with Listerine (LISTERINE® Cool Mint containing 21.6% alcohol) for 1 minute and to deliver the sputum in a tube afterwards. The DNA was extracted according to the Oragene Kit instructions and DNA in mouthwash samples was extracted by a standard salting-out procedure, as described by Miller et al. [42].

The analyses of DNA methylation were conducted by using next-generation sequencing (NGS). The method and quality control procedures are reported in detail in Moser et al. [43]. In short, 500 ng genomic DNA were bisulfite converted by using the *EZ DNA* Methylation-Gold Kit (Zymo Research, Irvine—USA). Afterwards, 1 μl of bisulfite-converted DNA was used for the first round PCR with the G2 Green Mix (Promega, Mannheim—Germany), which amplifies and tags the target regions. For bisulfite-specific primers, temperatures and concentrations see Table 2.

The quality of PCR-products was checked on a 1.5% agarose gel. In the second PCR, specific adapter primers bound to these tagged target sequences and additional DNA sequences were introduced to the PCR product, including barcode sequences, which were unique to a specific NGS primer pairs and were used for later identification of the samples. For the second PCR round, 1 μl of PCR products was used and quality control on a gel was conducted afterwards.

Then, 4 μl of the resulting PCD products were purified by Magnetic bead-based MagSi-NGS$^{PREP}$ Plus (MagnaMedics, Aachen; Germany). The dual-channel fluorometer (Quantus TM Fluorometer, Promega, Mannheim; Germany) was used for quantification and samples were adjusted for similar copy numbers for the NGS analysis. Finally, the paired-end sequencing was conducted by the Illumina MiSeq reagent Kit v2 (500 cycles- 2 x 250 paired end) and the MiSeq system (Illumina, San Diego; USA). These analyses were carried out in collaboration

**Table 2. Summary of amplicons characteristics, PCR-primers and PCR conditions (concentrations and annealing temperatures).**

| Name | Chromosomal location (bp) | Sequences forward (Fw) and reverse (Rv) primers | Number of CpGs | Primer concentrations/ annealing |
|---|---|---|---|---|
| *NR3C1 1$_F$* | chr5:142,783,541–142,783,911 (371) | Fw:CTTGCTTCCTGGCACGAGgggggtagatttggtttttt | 42 | 0.1μM/56.9˚C |
| | | Rv:CAGGAAACAGCTATGACtcccttccctaaaacctc | | |
| *FKBP5* | chr6:35,558,322–35,558,593 (272) | Fw:CTTGCTTCCTGGCACGAGttttgggttgaggatagaaagg | 5 | 0.1μM/56˚C |
| | | Rv: CAGGAAACAGCTATGACatccaaaacaactaacaaattctct | | |

with the BioChip Labor of the Center of Medical Biotechnology (ZMB, University Duisburg/ Essen).

The resulting FASTQ files were analyzed by the amplikyzer2 software [44,45] with default settings for DNA methylation analysis. Reads with less than 95% bisulfite conversion rate were excluded, as well as reads with less than 1000 reads per sample. Furthermore, CpG sites harboring SNPs were also excluded.

## Statistical analyses

For comparison of questionnaire data, Mann-Whitney-U tests were computed. For descriptive statistics of methylation data, mean methylation levels were calculated for NR3C1 and for FKBP5 bin 1, bin 2 and mean FKBP5, respectively. The methylation for each CpG was further mean-centered and tested for normal distribution. Since methylation data of NR3C1 $1_F$ promoter region and FKBP5 (intron 7) were not normally distributed, non-parametric methods were used. The mean methylation levels were additionally calculated for mean-centered data. In order to investigate the influence of group (BPD vs. healthy controls, in short HC) on CpG and mean methylation, Mann-Whitney-U tests were calculated for group comparisons of methylation levels. Bonferroni correction was applied for NR3C1 CpGs, resulting in $p = 0.05/42 = 0.0012$. For FKBP5, 5 bins (bin 1_1, bin 1_2; bin 2_1, bin 2_2, bin 2_3) were investigated, leading to $p = 0.05/5 = 0.01$. Spearman's rho correlation coefficients were calculated between averaged mean-centered methylation levels of FKBP5 bin 1, bin 2 and the whole FKBP5, as well as NR3C1 mean, and questionnaire results for the whole group. Spearman's rho correlation coefficients were also calculated for correlations between questionnaires (see Supplemental material). Bonferroni corrections were applied due to multiple testing (for correction thresholds see Tables 4 and 5). All statistical analyses were performed using the software SPSS version 25 (IBM Corp. Released 2017. IBM SPSS Statistics for Windows, Version 25.0. Armonk, NY: IBM Corp.). Post-hoc power analyses for the group differences in methylation were conducted by using the software G*Power Version 3.1.9.4 [46].

## Results

### Questionnaires

Table 3 shows the results of CTQ, IRI and SCl-90-R questionnaires and the comparisons between groups. Patients with BPD reported more severe psychopathological symptoms (SCL), increased distress (personal distress IRI) and lower perspective taking abilities (IRI) and more or severe adverse experiences during childhood.

### Epigenetics

Comparisons between groups for NR3C1 CpGs revealed differences for distinct CpGs, which did not survive correction for multiple comparisons (see S1 Table). Mean methylation levels averaged for all CpGs were lower in patients with BPD compared to the HC group (BPD: % methylation $M = 0.66\%$, $SD = 0.17$; mean-centered $M = -0.08$, $SD = 0.22$; mean rank = 37.87; HC: % methylation $M = 0.74\%$, $SD = 0.14$; mean centered $M = 0.08$, $SD = 0.29$; mean rank = 52.30; $U = 669.0$, $p = 0.008$ with effect size $d = 0.583$). Because of the overall low NR3C1 methylation, the difference between groups was small (i.e. 0.08%), Post-hoc power analysis for this comparison showed, however, that the statistical power $(1-\beta = 0.757)$ as determined using G*Power analysis was sufficiently large. Regarding FKBP5, there was no significant difference between groups (FKBP5 mean methylation BPD: % methylation $M = 84.38\%$, $SD = 2.48$; mean-centered $M = 0.09$, $SD = 0.73$; mean rank = 47.41; HC: % methylation $M = 83.86\%$,

**Table 3. Results of CTQ (n = 89), IRI (n = 89) and SCl-90-R (n = 86) questionnaires for the groups of patients with BPD and control participants.**

| | BPD | | HC | | Mann-Whitney-U-test | | |
|---|---|---|---|---|---|---|---|
| | *M* | *SD* | *M* | *SD* | *U* | *z* | *p* |
| Age | 26.3 | 5.7 | 24.0 | 3.1 | 783.5 | -1.70 | 0.089 |
| **Childhood Trauma Questionnaire** | | | | | | | |
| Total score | 66.0 | 19.9 | 33.2 | 9.8 | 130.5 | -7.06 | **<0.001** |
| Emotional abuse | 18.0 | 5.7 | 7.5 | 3.3 | 151.5 | -6.92 | **<0.001** |
| Physical abuse | 9.6 | 4.4 | 5.7 | 1.6 | 341.5 | -5.59 | **<0.001** |
| Sexual abuse | 10.5 | 6.0 | 5.4 | 1.5 | 404.5 | -5.41 | **<0.001** |
| Emotional neglect | 17.6 | 5.9 | 8.4 | 3.3 | 229.5 | -6.26 | **<0.001** |
| Physical neglect | 10.4 | 4.5 | 6.3 | 1.9 | 382.5 | -5.11 | **<0.001** |
| **Interpersonal Reactivity Index** | | | | | | | |
| Perspective taking | 13.0 | 6.9 | 19.0 | 4.0 | 452.5 | -4.42 | **<0.001** |
| Fantasy | 17.0 | 6.9 | 18.9 | 5.0 | 546.5 | -1.18 | 0.238 |
| Empathic concern | 18.9 | 6.3 | 19.9 | 4.1 | 976.0 | -0.12 | 0.908 |
| Personal distress | 21.8 | 4.6 | 12.7 | 4.2 | 157.0 | -6.85 | **<0.001** |
| **Symptom-Checklist 90-R** | | | | | | | |
| Somatization | 69.5 | 10.3 | 52.4 | 15.4 | 361.5 | -4.89 | **<0.001** |
| Obsessive-compulsiveness | 75.1 | 7.1 | 51.1 | 13.4 | 134.0 | -6.94 | **<0.001** |
| Social insecurity | 74.8 | 8.3 | 51.7 | 13.7 | 157.5 | -6.76 | **<0.001** |
| Depression | 75.9 | 5.8 | 51.2 | 14.4 | 75.5 | -7.42 | **<0.001** |
| Anxiety | 74.7 | 7.6 | 49.3 | 12.1 | 104.5 | -7.23 | **<0.001** |
| Aggression | 73.8 | 8.9 | 51.1 | 12.8 | 149.0 | -6.79 | **<0.001** |
| Phobic anxiety | 70.8 | 11.2 | 48.9 | 10.8 | 182.0 | -6.50 | **<0.001** |
| Paranoid thinking | 70.9 | 10.0 | 51.3 | 12.6 | 245.0 | -5.90 | **<0.001** |
| Psychoticism | 71.7 | 8.2 | 49.4 | 12.6 | 186.0 | -6.43 | **<0.001** |
| GSI | 79.1 | 3.4 | 57.8 | 16.6 | 206.5 | -6.78 | **<0.001** |
| PST | 79.8 | 1.1 | 58.2 | 19.5 | 309.5 | -6.24 | **<0.001** |
| PSDI | 74.5 | 6.0 | 57.2 | 12.1 | 196.0 | -6.36 | **<0.001** |

Mean (*M*) and standard deviation (*SD*) and results of Mann-Whitney-U Tests comparisons are shown.

*SD* = 2.46; mean-centered *M* = -0.09, *SD* = 0.76; mean rank = 41.59, *U* = 840.0, *p* = 0.285 with effect size *d* = 0.229; FKBP5 bin 1: BPD: *M* = 80.36%, *SD* = 3.47, mean rank = 45.36; HC: *M* = 80.49%, *SD* = 2.60, mean rank = 43.36, *U* = 930.0, *p* = 0.751; FKBP5 bin 2 BPD: *M* = 87.05%, *SD* = 2.43, mean rank = 48.80; HC: *M* = 86.10%, *SD* = 2.83, mean rank = 40.20, *U* = 779.0, *p* = 0.115; see also S2 Table). Post-hoc power analysis for group comparison of FKBP5 mean methylation expectedly revealed a low statistical power of 1-β = 0.179.

## Correlations

To explore possible associations of childhood trauma, psychopathology and empathy with each other, we calculated correlations between questionnaires. As expected, significant positive correlations between SCL-90 results and childhood trauma were found, showing that maltreatment during childhood was associated with increased psychopathological symptom load (S3 Table; all *p*'s ≤ 0.001). Psychopathological symptom severity was further associated with empathic perspective taking (inverse correlations with social insecurity, depression and global scores) and personal distress (positive correlations with all scores; all *p*'s ≤ 0.001). As already

**Table 4. Correlations (*r* (*p*)) of FKBP5 (bin 1 mean, bin 2 mean, mean bin 1 and 2) and NR3C1 methylation with psychopathological symptoms.**

|  | FKBP5 bin 1 | FKBP5 bin 2 | FKBP5 mean | NR3C1 mean |
|---|---|---|---|---|
| **Somatization** | 0.121 (0.271) | **0.269 (0.013)** | 0.205 (0.060) | -0.179 (0.099) |
| **Obsessive-compulsiveness** | 0.169 (0.121) | **0.243 (0.025)** | **0.228 (0.036)** | -0.162 (0.137) |
| **Social insecurity** | 0.135 (0.220) | **0.234 (0.031)** | 0.194 (0.075) | -0.152 (0.163) |
| **Depression** | 0.111 (0.314) | **0.249 (0.021)** | 0.201 (0.065) | -0.156 (0.150) |
| **Anxiety** | 0.110 (0.318) | **0.338* (0.002)** | **0.268 (0.013)** | **-0.271 (0.012)** |
| **Aggression** | 0.083 (0.450) | 0.191 (0.079) | 0.151 (0.167) | -0.156 (0.152) |
| **Phobic anxiety** | 0.132 (0.227) | **0.252 (0.020)** | 0.200 (0.066) | -0.142 (0.191) |
| **Paranoid thinking** | 0.080 (0.469) | **0.217 (0.046)** | 0.173 (0.122) | -0.161 (0.139) |
| **Psychoticism** | 0.068 (0.534) | **0.230 (0.034)** | 0.177 (0.105) | -0.191 (0.078) |
| **GSI** | **0.223 (0.041)** | **0.352* (0.001)** | **0.328* (0.002)** | -0.133 (0.224) |
| **PST** | 0.140 (0.202) | **0.273 (0.011)** | **0.244 (0.024)** | -0.158 (0.145) |
| **PSDI** | 0.139 (0.205) | **0.300 (0.005)** | **0.242 (0.026)** | -0.089 (0.413) |

Bonferroni correction for SCL for each region/bin: 0.05/12 = 0.0042*.

Significant correlations are printed in bold, correlation surviving correction due to multiple correction are marked with *.

reported in our previous studies (e.g., [35]), childhood trauma, especially emotional abuse, correlated with low perspective taking and high personal distress correlated with all CTQ scores (see also S4 Table; all p's ≤ 0.001).

Additional correlations were calculated between FKBP5 and NR3C1 methylation and psychopathology. Here, a positive correlation was found between FKBP5 methylation (bin 2) and anxiety symptoms. Similarly, positive correlations emerged between FKBP5 (general mean and bin 2) and the global severity index (GSI; Table 4). The negative correlation found between NR3C1 methylation and anxiety did not survive correction for multiple comparisons.

Moreover, significant correlations emerged between self-rated empathy and methylation, with high FKBP5 methylation being inversely correlated with perspective taking (correlations with bin 2 and mean) and fantasy (correlation with bin 2; Table 5). Correlations between FKBP5 methylation and CTQ scores did not reach statistical significance, correlations of CTQ with NR3C1 did not survive correction for multiple testing (see S5 Table).

## Discussion

The present study aimed to explore the methylation patterns of genes involved in the regulation of the glucocorticoid-driven stress response. Specifically, we were interested in the methylation of the FKBP5 intron 7 and NR3C1 $1_F$ promoter region in a clinical sample of patients with BPD in comparison to psychologically healthy controls. We further sought to examine

**Table 5. Correlations (*r* (*p*)) of FKBP5 (bin 1 mean, bin 2 mean, mean bin 1 and 2) and NR3C1 methylation levels with empathy (IRI).**

|  | FKBP5 bin 1 | FKBP5 bin 2 | FKBP5 mean | NR3C1 mean |
|---|---|---|---|---|
| **Perspective taking** | -0.202 (0.059) | **-0.335* (0.001)** | **-0.294* (0.005)** | 0.075 (0.487) |
| **Fantasy** | -0.068 (0.528) | **-0.266* (0.012)** | -0.174 (0.105) | 0.105 (0.329) |
| **Empathic concern** | -0.108 (0.317) | -0.185 (0.084) | -0.142 (0.186) | 0.085 (0.427) |
| **Personal distress** | 0.115 (0.287) | 0.055 (0.611) | 0.091 (0.398) | -0.197 (0.064) |

Bonferroni correction for IRI for each region/bin: 0.05/4 = 0.0125*.

Significant correlations are printed in bold, correlation surviving correction due to multiple correction are marked with *.

possible correlations of DNA methylation with childhood maltreatment, psychopathological symptoms and empathic abilities.

With regard to NR3C1, we found lower methylation in individuals with BPD compared with healthy subjects. However, the overall degree of methylation was small, such that a group difference of 0.08% needs to be critically discussed in terms of its biological relevance, especially in light of methodological limitations due a detection threshold of 0.46% [43]. However, post-hoc power analysis showed that the statistical power of 0.757 was fairly large. Notably, NR3C1 methylation further correlated inversely with childhood trauma, particularly emotional abuse, which just failed to reach significance after correction for multiple testing. Previous studies have reported both, increased methylation in traumatized individuals, as well as decreased methylation, depending on the specific CpG sites investigated (for reviews, see [27,28,47–50]. Moreover, there is some evidence to suggest that childhood trauma and stress-related psychopathology is related to the magnitude of NR3C1 (exon $1_F$) methylation [22,26,29]. Along similar lines, aberrant NR3C1 methylation may also play a role in the execution of acute stress responses [51–53].

When looking at the chaperone FKBP5, the evidence for a possible association of methylation patterns with maltreatment is even less straightforward, with some studies showing associations of childhood adversity with FKBP5 methylation [19,33,54], while others reporting negative results [55,56]. Such divergent findings may, in part, reside in the fact that the type of traumatic experience needs to be considered, i.e., early versus late, single incident versus repetitive traumatization, etc. [33]. In our study, even though patients with BPD reported high levels of childhood trauma, no association with FKBP5 methylation was found. However, high levels of personal distress (IRI) and trauma experiences correlated with the severity of psychopathology. Put another way, the experience of traumatic events during childhood is a risk-factor for psychopathology later in life. What is less clear to date is whether traumatic experiences during childhood leave an epigenetic mark on the stress system, and how pervasive such effect might be. Interestingly, in our study FKBP5 bin 2 methylation correlated with symptom severity, especially anxiety, as well as inversely with empathic perspective taking. This is consistent with previous reports showing increased FKBP5 methylation being associated with high levels of anxiety, depression and PTSD symptoms [31,54,57] (for review see [49]). Our findings expand on this indicating that FKBP5 methylation is also linked to empathy, which is compatible with studies demonstrating that emotional empathy is negatively affected by the experience of acute stress [58]. Our results show in addition that early stress may also affect empathy, and that methylation of stress-related genes such as FKBP5 could be a biological correlate of this finding.

Our study has several limitations. One concerns the low degree of methylation of the NR3C1 gene and the lack of group differences in FKBP5 methylation, as discussed above. The low NR3C1 levels in the present study are consistent with previous work reporting similarly low levels of NR3C1 methylation in relation to the experience of trauma (e.g. [23,52,59,60]. We believe that studies reporting methylation patterns beneath a certain methodological detection threshold need to be interpreted with caution and warrant replication [17,43,61–63]. Another shortcoming of the present study relates to the sample, which was exclusively female. A third limitation concerns the lack of investigation of polymorphic variation of the receptor coding gene [64], since the impact of FKBP5 methylation on psychopathology has been reported to be related to SNP variants within FKBP5 (e.g. [56,65]). Fourth, it needs to be taken into consideration that childhood trauma, psychopathology and empathy measures relied on self-report. Finally, the group of patients received psychopharmacological and psychotherapeutic therapy, which might have affected DNA-methylation.

In a more general vein, DNA methylation is one mechanism for epigenetic modification of gene expression that probably evolved as intermediate-term responses to adjust to environmental conditions including exposure to stress [66]. That is, epigenetic responses to stressful conditions may allow individuals to function better under the stress, but may be associated with long-term negative consequences. However, it can be assumed that there are large inter-individual differences in stress responsivity and in causes of stress, especially considering that modern stressful conditions are variable and different from the stressful conditions during our evolutionary history. A related point is that control of glucocorticoid function can be expected to be under complex influences that may defy simple linear cause/effect hypotheses.

In summary, the present study demonstrates that FKBP5 (bin2) hypermethylation was associated with psychopathology and deficits in empathy. Contrary to predictions, no effect of childhood maltreatment on DNA methylation was observed. Furthermore, no difference in methylation emerged between a clinical and a non-clinical group regarding FKBP5, yet a small, but biologically questionable difference occurred regarding the average degree of NR3C1 ($1_F$) methylation. Future research may need to control for polymorphic variation of genes involved in stress regulation. In addition, the specific nature of stressful life-events needs to be taken into consideration more strictly, as well as the developmental timepoint of traumatization.

## Supporting information

**S1 Table. Means (*M*) and standard deviations (*SD*) of NR3C1(exon $1_F$) methylation levels in absolute (%), mean-centered and ranked data for patients with BPD (n = 45) and HC (n = 44).** Group differences were calculated by Mann-Whitney-U tests (*U*, *p*); significant differences are marked in bold. These differences would did not survive correction due to multiple testing. Bonferroni correction for $p = 0.05/42 = 0.0012$.
(DOCX)

**S2 Table. Means (*M*) and standard deviations (*SD*) of FKBP5 methylation levels in absolute, mean-centered and ranked data for patients with BPD (n = 44) and HC (n = 44).** Group differences were calculated by Mann-Whitney-U tests (*U*, *p*).
(DOCX)

**S3 Table. Spearman's rho correlation coefficients (r (p)) calculated between psychopathological symptoms (SCL-90-R), childhood trauma (CTQ) and empathy (IRI).** Bonferroni correction for SCL scales and CTQ: $0.05/72 = 0.00042^*$; and IRI: $0.05/48 = 0.0010^*$; Therefore, only correlations with $p < 0.001$ were considered as significant.
(DOCX)

**S4 Table. Correlations (Spearman (r (p))) between childhood trauma scales (CTQ) and empathy (IRI).** Bonferroni correction for IRI scales: $0.05/24 = 0.0021^*$.
(DOCX)

**S5 Table. Correlations (Spearman (r (p))) between childhood trauma scales (CTQ) and FKBP5 (bin 1 mean, bin 2 mean, mean bin 1 and 2) and NR3C1 mean methylation for patients with BPD and HC (n = 88).** Significant correlations are printed in bold, however these correlation did not survive correction due to multiple testing. Bonferroni correction for IRI for each region/bin: $0.05/6 = 0.0083^*$.
(DOCX)

**S6 Table. Correlations between IRI scales.**
(DOCX)

## Acknowledgments

We gratefully acknowledge the technical support of the Department of Genetic Psychology, Faculty of Psychology, Ruhr-University Bochum (Bochum, Germany; led by Professor Robert Kumsta). We also thank Dr. Jasmin Beygo from the Institute of Human Genetics, University Hospital Essen, University Duisburg-Essen and Priv.-Doz. Dr. Ludger Klein-Hitpass from the Institute of Cell Biology (Tumor Research), University Hospital Essen, University of Duisburg-Essen for conducting NGS and for their support in analyzing the data. We would also like to thank Professor Robert Kumsta and Dr. Dirk Moser (Department of Genetic Psychology, Faculty of Psychology, Ruhr-University Bochum, Germany) for their helpful comments on an earlier draft of the manuscript.

## Author Contributions

**Conceptualization:** Vera Flasbeck, Martin Brüne.

**Data curation:** Vera Flasbeck.

**Formal analysis:** Vera Flasbeck, Martin Brüne.

**Investigation:** Vera Flasbeck.

**Methodology:** Vera Flasbeck.

**Project administration:** Martin Brüne.

**Resources:** Martin Brüne.

**Supervision:** Martin Brüne.

**Writing – original draft:** Vera Flasbeck.

**Writing – review & editing:** Martin Brüne.

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
