## [Decision Letter · Decision Letter 0]

20 Jan 2021

PONE-D-20-32998

Association between childhood maltreatment, psychopathology and DNA methylation of genes involved in stress regulation: Evidence from a study in Borderline Personality Disorder

PLOS ONE

Dear Dr. Brüne,

Thank you for submitting your manuscript to PLOS ONE. After careful consideration, we feel that it has merit but does not fully meet PLOS ONE’s publication criteria as it currently stands. Therefore, we invite you to submit a revised version of the manuscript that addresses the points raised during the review process.

We look forward to receiving your revised manuscript.

Kind regards,

Alexandra Kavushansky, PhD

Academic Editor

PLOS ONE

Reviewers' comments:

Reviewer's Responses to Questions

**Comments to the Author**

1. Is the manuscript technically sound, and do the data support the conclusions?

Reviewer #1: Yes

Reviewer #2: Yes

2. Has the statistical analysis been performed appropriately and rigorously? 

Reviewer #1: Yes

Reviewer #2: Yes

3. Have the authors made all data underlying the findings in their manuscript fully available?

Reviewer #1: No

Reviewer #2: Yes

4. Is the manuscript presented in an intelligible fashion and written in standard English?

Reviewer #1: Yes

Reviewer #2: Yes

5. Review Comments to the Author

Reviewer #1: In this manuscript, authors investigated the association between NR3C1 and FKBP5 methylation levels and BPD. They also tested whether FKBP5 methylation correlated with childhood trauma, psychopathology and empathy. Those research questions are highly significant for the field. Their hypothesis is really strong and supported by previous literature, which authors summarized very well. The statistical analyses are robust and well defined. Even though the sample size of 89 (44 healthy participants and 45 patients diagnosed with BPD) is not very impressive, it is sufficient for a candidate gene study. However, authors should address some issues to improve the manuscript.

Major issues:

Epigenetics Analysis between Groups:

- Authors should perform a post-hoc power analysis for NR3C1 and FKBP5 methylation analyses to report the lowest difference between groups that they are able to identify given their sample size and p-value thresholds.

- I would like to see the statistics (mean, sd, p values) for all 42 NR3C1 CpGs and 5 FKBP5 CpGs as a supplementary table.

Correlations:

- Even though the correlations between FKBP5 and CTQ scores did not reach the statistical significance, I would like to see the correlations as a supplementary table.

- Is there a reason why authors did not perform the correlation analyses for NR3C1? If so, they should state their rationale. If not, I really would like to see the correlations of NR3C1 methylation with psychopathological symptoms, IRI and CTQ.

Minor issues:

- Missing parentheses at line 68.

- At line 90, “more pronounced methylation” can be understood as hypermethylation, which is not always the case for both genes, since in the intro authors state that “depending on the CpG site, previous studies reported hypo-or hypermethylation in association with childhood maltreatment”. It would be better to use another term (such as changes in methylation levels) instead of “more pronounced methylation”.

- Line 175, “CpG sites created by SNPs” is not the correct use, assuming that authors mean “CpG sites harboring SNPs”.

- Authors should introduce the HC abbreviation in their first use.

- Table 4: There is a typo for the r value for the correlation between OCD and FKBP5 bin 2. 0,243 should be 0.243

- Table 5 is not referenced in text. I assume it should be referenced at line 244.

Reviewer #2: This paper provides a significant contribution to the understanding of manifestations that can be associated with borderline personality and other psychiatric illnesses. I think that can be accepted virtually as is, but I suggest the following revisions.

One general comment is that the research could be placed in its adaptive context, which may shed some light on the reasons for some of the paradoxes documented in this study and inconsistencies in the existing spectrum of studies. In particular, mechanisms for epigenetic modification probably evolved as intermediate-term responses to adjust to environmental conditions (Ewald and Swain Ewald 2019). Epigenetic responses to stressful conditions may allow individuals to function better under the stress, but may be associated with long-term negative consequences. But there is no reason to think that these responses would be uniform across individuals and stressful conditions, especially considering that modern stressful conditions are variable and different from the stressful conditions during our evolutionary history. A related point is that control of glucocorticoid function can be expected to be under complex influences that may defy simple linear cause/effect hypotheses. I think that these points could be dealt with succinctly in an additional paragraph at the end of the Discussion.

I would also like to see the authors provide a line in the Abstract that directly addresses the findings in the context of the structure of their controlled study. Namely that differences between BPD patients and control were statistically significant but with small effect size. In this regard I think that the authors are being too dismissive of their findings in the body of the paper when they state on lines 213-214, “…because of overall low NR3C1 methylation, the difference between groups was negligibly small (i.e. 0.08 %), which is not considered biologically meaningful...” The findings could be biologically meaningful but explain only a tiny amount of the variation. It might be that the categorization of patients into BPD and control groups is reflects so much variation in causal risk factors and in psychiatric manifestations that the measured epigenetic changes explain only a small amount of the variation in the causal mechanisms of BPD relative to controls. These complications could also help explain the variation in findings in the literature that the authors mention in the Introduction.

Minor comments:

Line 65: Semantics: I suggest “In recent years” instead of “In the last years”

Line 144: Greater specificity is needed with regard to the particular variety of Listerine that was used.

6. PLOS authors have the option to publish the peer review history of their article (what does this mean?). If published, this will include your full peer review and any attached files.

Reviewer #1: No

Reviewer #2: No

---

## [Author Response · Author response to Decision Letter 0]

3 Feb 2021

Reviewer #1: 

In this manuscript, authors investigated the association between NR3C1 and FKBP5 methylation levels and BPD. They also tested whether FKBP5 methylation correlated with childhood trauma, psychopathology and empathy. Those research questions are highly significant for the field. Their hypothesis is really strong and supported by previous literature, which authors summarized very well. The statistical analyses are robust and well defined. Even though the sample size of 89 (44 healthy participants and 45 patients diagnosed with BPD) is not very impressive, it is sufficient for a candidate gene study. However, authors should address some issues to improve the manuscript.

Response: Many thanks, we are grateful for the reviewer’s encouraging comments.

Major issues:

Epigenetics Analysis between Groups:

- Authors should perform a post-hoc power analysis for NR3C1 and FKBP5 methylation analyses to report the lowest difference between groups that they are able to identify given their sample size and p-value thresholds.

Response: Thank you very much for this excellent suggestions. We inserted power analyses for group differences to the results section. Therefore, Cohen’s d was calculated as effect sizes. It now reads: 

Line 197-199: “Post-hoc power analyses for the group comparisons of methylation levels were conducted by using the software G*Power Version 3.1.9.4 [46]. 

Line 218-221 “Because of overall low NR3C1 methylation, the difference between groups was small (i.e. 0.08 %). Post-hoc power analysis for this comparison showed, however, that the statistical power (1-β = 0.757) as determined using G*Power analysis was sufficiently large.”

[…] line .228-230: “Post-hoc power analysis for group comparison of FKBP5 mean methylation expectedly revealed a low statistical power of 1-β = 0.179.”

Discussion, Line 278: “However, post-hoc power analysis showed that the statistical power of 0.757 was fairly large.”

- I would like to see the statistics (mean, sd, p values) for all 42 NR3C1 CpGs and 5 FKBP5 CpGs as a supplementary table.

Response: We added the statistics for all CpGs to the supplemental information (S1 and S2 tables.

Correlations:

- Even though the correlations between FKBP5 and CTQ scores did not reach the statistical significance, I would like to see the correlations as a supplementary table.

Response: We agree with the reviewer that this information is important to present. We inserted the FKBP5-CTQ correlations, as well as NR3C1-CTQ correlations to the supplemental information (S5 Table). 

Line 257-259: “Correlations between FKBP5 methylation and CTQ scores did not reach statistical significance, correlations of CTQ with NR3C1 did not survive correction for multiple testing (see S5 Table).” 

S5 Table. Correlations (Spearman (r (p)) between childhood trauma scales (CTQ) and FKBP5 (bin 1 mean, bin 2 mean, mean bin 1 and 2) and NR3C1 mean methylation for patients with BPD and HC (n = 88). Significant correlations are printed in bold, however these correlations did not survive correction for multiple testing.

Childhood Trauma Questionnaire FKBP5 bin 1 FKBP5 bin 2 FKBP5 mean NR3C1 mean

Emotional abuse 0.023 (0.830) -0.055 (0.609) -0.041 (0.705) -0.267 (0.011)

Physical abuse -0.024 (0.824) -0.073 (0.501) 0.038 (0.722) -0.251 (0.018)

Sexual abuse -0.136 (0.207) 0.108 (0.318) -0.002 (0.988) -0.128 (0.232)

Emotional neglect 0.045 (0.677) 0.057 (0.601) 0.049 (0.650) -0.200 (0.060)

Physical neglect 0.144 (0.182) 0.139 (0.197) 0.144 (0.180) -0.233 (0.028)

Total score 0.021 (0.848) 0.110 (0.306) 0.074 (0.491) -0.226 (0.033)

Bonferroni correction for IRI for each region/bin: 0.05/ 6 = 0.0083*

- Is there a reason why authors did not perform the correlation analyses for NR3C1? If so, they should state their rationale. If not, I really would like to see the correlations of NR3C1 methylation with psychopathological symptoms, IRI and CTQ.

Response: In the first version of the manuscript, we refrained from focusing elaborately on NR3C1 results, because of overall low NR3C1 methylation levels. However, as noted by reviewer 2, in the revised version, we reported NR3C1 data, but discussed the meaning of low levels of methylation in the discussion section. We therefore also added correlation with NR3C1 to Tables 4 and 5 and S5 Table. 

Minor issues:

- Missing parentheses at line 68.

Response: Thanks, we added the parentheses.

- At line 90, “more pronounced methylation” can be understood as hypermethylation, which is not always the case for both genes, since in the intro authors state that “depending on the CpG site, previous studies reported hypo-or hypermethylation in association with childhood maltreatment”. It would be better to use another term (such as changes in methylation levels) instead of “more pronounced methylation”.

Response: We thank the reviewer for this advice and revised this sentence accordingly: Line 92: “We hypothesized changes in methylation levels of both genes in BPD.”

- Line 175, “CpG sites created by SNPs” is not the correct use, assuming that authors mean “CpG sites harboring SNPs”.

Response: We corrected the phrase according to the reviewer’s recommendation: Line 176-177: “Furthermore, CpG sites harboring SNPs were also excluded.”

- Authors should introduce the HC abbreviation in their first use.

Response: Thank you, the abbreviation is introduced in the revised version. 

- Table 4: There is a typo for the r value for the correlation between OCD and FKBP5 bin 2. 0,243 should be 0.243

Response: Thanks, this typo was corrected

- Table 5 is not referenced in text. I assume it should be referenced at line 244.

Response: we added the reference for Table 5 at line 257. 

Reviewer #2: 

This paper provides a significant contribution to the understanding of manifestations that can be associated with borderline personality and other psychiatric illnesses. I think that can be accepted virtually as is, but I suggest the following revisions.

One general comment is that the research could be placed in its adaptive context, which may shed some light on the reasons for some of the paradoxes documented in this study and inconsistencies in the existing spectrum of studies. In particular, mechanisms for epigenetic modification probably evolved as intermediate-term responses to adjust to environmental conditions (Ewald and Swain Ewald 2019). Epigenetic responses to stressful conditions may allow individuals to function better under the stress, but may be associated with long-term negative consequences. But there is no reason to think that these responses would be uniform across individuals and stressful conditions, especially considering that modern stressful conditions are variable and different from the stressful conditions during our evolutionary history. A related point is that control of glucocorticoid function can be expected to be under complex influences that may defy simple linear cause/effect hypotheses. I think that these points could be dealt with succinctly in an additional paragraph at the end of the Discussion.

I would also like to see the authors provide a line in the Abstract that directly addresses the findings in the context of the structure of their controlled study. Namely that differences between BPD patients and control were statistically significant but with small effect size. In this regard I think that the authors are being too dismissive of their findings in the body of the paper when they state on lines 213-214, “…because of overall low NR3C1 methylation, the difference between groups was negligibly small (i.e. 0.08 %), which is not considered biologically meaningful...” The findings could be biologically meaningful but explain only a tiny amount of the variation. It might be that the categorization of patients into BPD and control groups is reflects so much variation in causal risk factors and in psychiatric manifestations that the measured epigenetic changes explain only a small amount of the variation in the causal mechanisms of BPD relative to controls. These complications could also help explain the variation in findings in the literature that the authors mention in the Introduction.

Response: We are extremely grateful to the reviewer for his or her encouraging and supportive comments that helped us improve the manuscript. In detail, we added NR3C1 findings to the abstract and discussed the findings as recommended. As also suggested by Reviewer 1, the NR3C1 results were presented for correlations with questionnaires and data for all CpGs were added to the supplemental information. 

Discussion: Line 274-280: “With regard to NR3C1, we found lower methylation in individuals with BPD compared with healthy subjects. However, the overall degree of methylation was small, such that a group difference of 0.08% needs to be critically discussed in terms of its biological relevance, especially in light of methodological limitations due a detection threshold of 0.46 % [43]. However, post-hoc power analysis showed that the statistical power of 0.757 was fairly large. Notably, NR3C1 methylation further correlated inversely with childhood trauma, particularly emotional abuse, which just failed to reach significance after correction for multiple testing. […] 

Line 321-330: “In a more general vein, DNA methylation is one mechanism for epigenetic modification of gene expression that probably evolved as intermediate-term responses to adjust to environmental conditions including exposure to stress [66]. That is, epigenetic responses to stressful conditions may allow individuals to function better under the stress, but may be associated with long-term negative consequences. However, it can be assumed that there are large inter-individual differences in stress responsivity and in causes of stress, especially considering that modern stressful conditions are variable and different from the stressful conditions during our evolutionary history. A related point is that control of glucocorticoid function can be expected to be under complex influences that may defy simple linear cause/effect hypotheses.” […]

Line 333-336: “Furthermore, no difference in methylation emerged between a clinical and a non-clinical group regarding FKBP5, yet a small, but biologically questionable difference occurred regarding the average degree of NR3C1 (1F) methylation.”

Abstract Line 35-37: “Methylation of NR3C1 was lower in BPD patients compared to controls, yet with small differences. The results are discussed regarding their biological relevance, including possible evolutionary explanations.”

Minor comments:

Line 65: Semantics: I suggest “In recent years” instead of “In the last years”

Response: Thank you, we changed this phrase according to the reviewer’s suggestion. 

Line 144: Greater specificity is needed with regard to the particular variety of Listerine that was used.

Response: We added the following information for the listerine sampling: Line 144-146: “For mouthwash samples, participants were asked to rinse the mouth with Listerine (LISTERINE® Cool Mint from Johnson & Johnson GmbH, containing 21.6 % alcohol) for 1 minute and to spit the content in a tube afterwards.”

---

## [Decision Letter · Decision Letter 1]

1 Mar 2021

Association between childhood maltreatment, psychopathology and DNA methylation of genes involved in stress regulation: Evidence from a study in Borderline Personality Disorder

PONE-D-20-32998R1

Dear Dr. Brüne,

We’re pleased to inform you that your manuscript has been judged scientifically suitable for publication and will be formally accepted for publication once it meets all outstanding technical requirements.

Kind regards,

Alexandra Kavushansky, PhD

Academic Editor

PLOS ONE

Additional Editor Comments (optional):

Reviewers' comments:

Reviewer's Responses to Questions

**Comments to the Author**

1. If the authors have adequately addressed your comments raised in a previous round of review and you feel that this manuscript is now acceptable for publication, you may indicate that here to bypass the “Comments to the Author” section, enter your conflict of interest statement in the “Confidential to Editor” section, and submit your "Accept" recommendation.

Reviewer #1: All comments have been addressed

Reviewer #2: All comments have been addressed

2. Is the manuscript technically sound, and do the data support the conclusions?

Reviewer #1: Yes

Reviewer #2: Yes

3. Has the statistical analysis been performed appropriately and rigorously? 

Reviewer #1: Yes

Reviewer #2: Yes

4. Have the authors made all data underlying the findings in their manuscript fully available?

Reviewer #1: Yes

Reviewer #2: Yes

5. Is the manuscript presented in an intelligible fashion and written in standard English?

Reviewer #1: Yes

Reviewer #2: Yes

6. Review Comments to the Author

Reviewer #1: The authors have addressed all the comments in detail. I do not have further comments and suggestions.

Reviewer #2: I think that this revised manuscript is now ready for publication in PLOS-One. The analysis is sound and the authors have appropriately dealt with my comments.

7. PLOS authors have the option to publish the peer review history of their article (what does this mean?). If published, this will include your full peer review and any attached files.

Reviewer #1: No

Reviewer #2: No

---

## [Editor Report · Acceptance letter]

3 Mar 2021

PONE-D-20-32998R1 

Association between childhood maltreatment, psychopathology and DNA methylation of genes involved in stress regulation: Evidence from a study in Borderline Personality Disorder. 

Dear Dr. Brüne:

I'm pleased to inform you that your manuscript has been deemed suitable for publication in PLOS ONE. Congratulations! Your manuscript is now with our production department. 

Kind regards, 

on behalf of

Dr. Alexandra Kavushansky 

Academic Editor

PLOS ONE